# AHEAD-OF-TIME P-TUNING

## ABSTRACT

This paper proposes a new parameter-efficient method for fine-tuning, AoT P-Tuning. This method adds input-dependent biases before evaluating the Transformer layer, reducing the required evaluation time while allowing multi-task inference with a single backbone model for evaluating different tasks in a single batch. We experimented with the proposed method on the GLUE and SuperGLUE benchmarking datasets using RoBERTa-Base, RoBERTa-Large, and DeBERTa-XL backbone models. Our findings show that AoT P-tuning performed on par with or better than P-Tuning v2 and comparable to other baselines for efficient fine-tuning while being faster during inference.

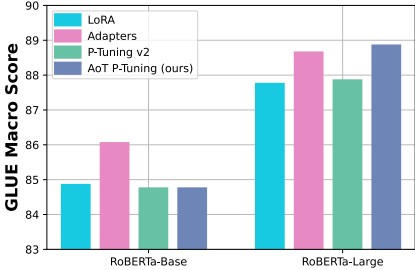 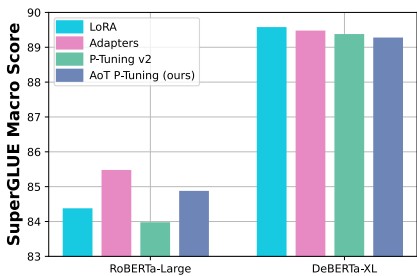

Figure 1: GLUE and SuperGLUE Macro scores for different backbone model scales. See Section 4.2 for more details.

## 1 INTRODUCTION

P-Tuning (Liu et al., 2021b;a; Lester et al., 2021) is a promising way to fine-tune large Language Models (LMs) (Devlin et al., 2019; Lan et al., 2020; Liu et al., 2019; Radford et al., 2019). While it currently underperforms compared to other methods for parameter-efficient fine-tuning (Hu et al., 2022; Houlsby et al., 2019) on a wide range of tasks (Ding et al., 2022), it has a practical, valuable property that allows it to evaluate different trained prompts parallel in a multi-task manner (i.e., a single backbone LM could be used for different tasks during inference, which can simplify model serving in real-world applications) (Lester et al., 2021). This property is why researchers aim to further develop P-Tuning methods.

Although it is possible to perform multi-task evaluation with P-Tuning, it introduces significant computational overhead due to the concatenation of prefixes to sequences and the evaluation of the attention mechanism (Vaswani et al., 2017) on longer sequences.

We propose a simple mechanism for parameter-efficient fine-tuning of Language Models, namely **Ahead-of-Time (AoT) P-Tuning**, for which we add input-dependent bias before each Transformer layer. Same as P-Tuning, it is possible to use AoT P-Tuning in multi-task inference setups when a single backbone LM is used for several downstream tasks.

The contributions of this paper can be summarized as follows:

1. We described the intuition behind AoT P-Tuning, which illustrates the connection of the proposed method with P-Tuning.

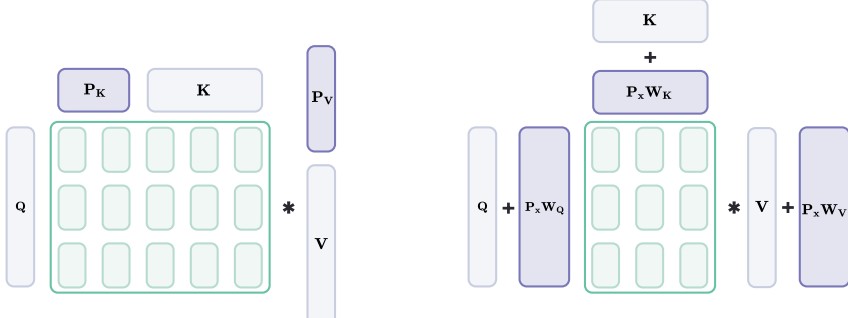

Figure 2: Schematic comparison of P-Tuning v2 (left), and AoT P-Tuning (right). Since the sequence length is not increased, AoT P-Tuning takes significantly less time to evaluate, only requiring the overhead of adding biases to the input sequence (See Section 4.3 for experiments with inference speed).

2. We proposed two reparameterizations of AoT P-Tuning weights: first based on a factorized matrix trained from scratch, and second based on a LM's embeddings matrix passed through a trainable Fully Connected network.

3. We experimented with the proposed method on GLUE, and SuperGLUE Benchmarking Datasets (Wang et al., 2018; 2019) with the RoBERTa (Liu et al., 2019) and DeBERTa (He et al., 2020) models and observed that AoT P-Tuning performed on par with or better than P-Tuning v2, comparable to other baselines for efficient fine-tuning while being faster than them.

## 2 RECENT WORKS

Currently, a wide range of different methods could be referenced with P-Tuning. Liu et al. (2021b) proposed to add soft prompts to the embeddings of GPT-2's input sequence (Radford et al., 2019) to train it on classification tasks. Lester et al. (2021) proposed a scheme similar to the one used in Liu et al. (2021b), but trained a T5 model (Raffel et al., 2020) with P-Tuning to show how the performance of the method changes with the increased scale of the backbone model.

Recently, Qin & Eisner (2021); Li & Liang (2021); Liu et al. (2021a) proposed to add prefixes not only to input embeddings but also at each layer of the Transformer model. In addition, Liu et al. (2021a) suggested training a linear classification head on top of the backbone model instead of utilizing a LM head to obtain classification results.

Due to this range of similar methods, we will follow the naming used by Liu et al. (2021a) and refer to Prompt-Tuning (adding soft prompts to the input embeddings) as P-Tuning v1 and to Prefix-Tuning (adding soft prefixes at each layer of Transformer backbone) as P-Tuning v2.

Hu et al. (2022) proposed to train low-rank changes of attention weights, while Houlsby et al. (2019) fine-tuned additional model layers, which can also be considered parameter-efficient. Ben Zaken et al. (2022) proposed to fine-tune only bias terms of the model.

## 3 AHEAD-OF-TIME P-TUNING

For readers' convenience, we provided background to Transformer evaluation and P-Tuning v1/v2 methods, which are relatable to the proposed method in Appendix Section A.

### 3.1 ON THE OVERHEAD OF RECENT METHODS

While the Transformer model has $\mathbb{O}(n^2)$ time complexity and GPU memory consumption for sequence length $n$. For P-Tuning v1, this complexity transforms into $\mathbb{O}((n + p)^2)$ since the length of input sequence is increased by the length of the prompt $p$, while for P-Tuning v2 the complexity is equal to $\mathbb{O}(n(n + p))$.

| Method | Parameter Efficient | Zero-Cost | Multi-Task Inference |
|---|---|---|---|
| Fine-Tuning | ✗ | ✓ | ✗ |
| LoRA | ✓ | ✗ | ✓ |
| LoRA Fused | ✓ | ✓ | ✗ |
| Adapters | ✓ | ✗ | ✓ |
| BitFit | ✓ | ✓ | ✓ |
| P-Tuning v1/v2 | ✓ | ✗ | ✓ |
| AoT P-Tuning (ours) | ✓ | ✓ | ✓ |

Table 1: Schematic comparison of recent fine-tuning methods with AoT P-Tuning. Recent fine-tuning approaches either allow inference with no computational overhead or multi-task inference. See Section 3.1 for details.

Liu et al. (2021a) showed that for some tasks, the prompt length $p$ could reach values of 100, increasing time and memory footprint during evaluation.

For the Adapters approach (Houlsby et al., 2019), the overhead appears since additional layers are injected into the computation of a model. However, it is possible to perform multi-task inference with it by passing the weights of these layers with the batch.

We could consider two usage scenarios for LoRA (Hu et al., 2022). For the first one, we fuse original weights with a trained low-rank addition. In such a case, we will see no inference speed overhead since the obtained model will have the same structure as the original pre-trained model. While it is theoretically possible to perform multi-task inference with a fused model similar to Adapters, doing so will require allocating a massive amount of GPU memory. E.g., for the DeBERTa-XL model with layers of hidden size $d = 1024$ and $l = 48$, passing a batch with $b$ sequences will require one to store $1024 * 1024 * 48 * 4 * b$ parameters to be given to the model, where 4 is number of parameter matrices of Attention module. It is easy to note that with $b = 4$, we will already exceed the number of original parameters in the model, which is impractical. To overcome this, it is possible to fuse LoRA weights to perform multi-task inference. However, doing so can introduce computational overhead.

See Table 1 for the schematic comparison of recent methods in terms of inference speed overhead and ability to perform multi-task inference.

### 3.2 PROPOSED MECHANISM

With AoT P-Tuning, we propose to augment each Transformer layer with a simple procedure. We define trainable matrices $\boldsymbol{P}^i \in \mathbb{R}^{|\boldsymbol{V}| \times d}$ for each layer. Then, before the evaluation of the $i$-th layer, we modify the hidden states as follows:

$$\boldsymbol{H}'^i = \boldsymbol{H}^i + \{\boldsymbol{P}^i_{x_1}, \dots, \boldsymbol{P}^i_{x_n}\} \in \mathbb{R}^{n \times d}, \tag{1}$$

where $\boldsymbol{P}^i_{x_j} \in \mathbb{R}^d$ is a lookup of $x_j$-th prompt embedding from $\boldsymbol{P}^i$. Such a scheme allows us to save a significant amount of time during evaluation since AoT P-Tuning does not imply an increase in sequence length. While $\boldsymbol{P}^i$ in naive implementation will require lot of memory to store parameters, in the following Section 3.3, we describe reparametrizations which make training more tractable.

Note that AoT P-Tuning, same as plain P-Tuning, could be evaluated in parallel with several tasks in a batch due to the fact that performing look-up from $\boldsymbol{P}$ can be easily parallelized. See Appendix Section B for explained intuition behind the proposed method.

As for P-Tuning v1 and P-Tuning v2, we only optimize parameters of $\boldsymbol{P}$ and Classification Head during fine-tuning.

### 3.3 ON THE PARAMETER EFFICIENCY OF AoT P-TUNING

It is notable that, in most cases, it is not feasible to optimize the weight $\boldsymbol{P} \in \mathbb{R}^{|\boldsymbol{V}| \times d}$ for each layer. If we consider training RoBERTa-Large with such a scheme (which has $|\boldsymbol{V}| = 50265$, $d = 1024$ and

| RoBERTa-Base | | | | |
|---|---|---|---|---|
| Model | STS-B | SST-2 | RTE | QQP | |
| Fine-Tuning | 90.6 ± 0.3 | 95.0 ± 0.2 | 81.2 ± 0.7 | 89.6 ± 0.2 | |
| Adapters | **90.7 ± 0.2** | 94.4 ± 0.3 | 80.5 ± 2.0 | **89.2 ± 0.1** | |
| LoRA | 90.1 ± 0.3 | 94.3 ± 0.5 | 80.5 ± 1.8 | 86.3 ± 0.3 | |
| BitFit | 90.3 ± 0.1 | 94.5 ± 0.5 | **80.9 ± 1.4** | 85.5 ± 0.6 | |
| P-Tuning v1 | 86.9 ± 0.9 | 94.0 ± 0.3 | 60.3 ± 2.4 | 82.2 ± 1.5 | |
| P-Tuning v2 | 89.2 ± 0.3 | **94.6 ± 0.2** | 80.5 ± 3.4 | 86.4 ± 3.3 | |
| Kron. AoT P-Tuning (ours) | 89.7 ± 0.2 | 94.0 ± 0.2 | 77.6 ± 1.4 | 88.2 ± 0.1 | |
| FC AoT P-Tuning (ours) | 90.0 ± 0.2 | 94.4 ± 0.3 | 78.0 ± 1.3 | 87.9 ± 0.2 | |
| | QNLI | MRPC | MNLI | CoLA | Macro |
| Fine-Tuning | 92.4 ± 0.1 | 90.8 ± 0.5 | 87.0 ± 0.3 | 63.8 ± 1.4 | 86.3 |
| Adapters | **92.4 ± 0.2** | **91.1 ± 1.1** | **86.8 ± 0.1** | **63.0 ± 1.3** | **86.0** |
| LoRA | 91.6 ± 0.3 | 90.6 ± 0.7 | 84.7 ± 0.3 | 60.3 ± 1.0 | 84.8 |
| BirFit | 90.9 ± 0.5 | 90.5 ± 1.7 | 85.0 ± 0.1 | 60.4 ± 1.2 | 84.7 |
| P-Tuning v1 | 88.3 ± 0.5 | 82.0 ± 1.7 | 80.8 ± 0.6 | 45.8 ± 27.1 | 77.5 |
| P-Tuning v2 | 91.9 ± 1.6 | 89.1 ± 1.1 | 85.3 ± 0.2 | 60.7 ± 2.6 | 84.7 |
| Kron. AoT P-Tuning (ours) | 90.7 ± 0.4 | 89.5 ± 1.1 | 84.6 ± 0.1 | 59.3 ± 1.2 | 84.2 |
| FC AoT P-Tuning (ours) | 91.3 ± 0.4 | 90.3 ± 0.3 | 85.4 ± 0.1 | 60.3 ± 2.2 | 84.7 |
| RoBERTa-Large | | | | | |
| Model | STS-B | SST-2 | RTE | QQP | |
| Fine-Tuning | 91.9 ± 0.2 | 96.1 ± 0.4 | 88.1 ± 1.5 | 90.3 ± 0.2 | |
| Adapters | **92.1 ± 0.2** | 96.3 ± 0.4 | **90.0 ± 0.1** | **94.2 ± 0.1** | |
| LoRA | 91.4 ± 0.3 | 95.9 ± 0.2 | 87.2 ± 18.7 | 93.7 ± 23.7 | |
| BitFit | 91.8 ± 0.2 | 96.2 ± 0.4 | 87.7 ± 0.8 | 87.2 ± 0.6 | |
| P-Tuning v1 | 75.5 ± 6.3 | 94.4 ± 0.4 | 62.8 ± 2.3 | 76.9 ± 2.5 | |
| P-Tuning v2 | 91.0 ± 0.4 | 96.1 ± 0.3 | 87.4 ± 1.5 | 86.6 ± 0.6 | |
| Kron. AoT P-Tuning (ours) | 91.1 ± 0.8 | 96.2 ± 0.2 | 84.8 ± 1.3 | 89.4 ± 0.1 | |
| FC AoT P-Tuning (ours) | 91.7 ± 0.4 | **96.7 ± 0.1** | 88.4 ± 0.9 | 88.7 ± 0.2 | |
| | QNLI | MRPC | MNLI | CoLA | Macro |
| Fine-Tuning | 94.3 ± 0.2 | 91.6 ± 0.6 | 89.9 ± 0.2 | 68.1 ± 1.9 | 88.8 |
| Adapters | 91.3 ± 0.4 | 90.1 ± 0.2 | 67.2 ± 1.3 | **87.7 ± 18.5** | 88.6 |
| LoRA | 91.0 ± 7.2 | 88.9 ± 24.0 | 66.3 ± 1.9 | 87.4 ± 1.6 | 87.7 |
| BitFit | 94.1 ± 0.4 | 91.0 ± 1.0 | 89.4 ± 0.1 | 69.8 ± 3.1 | 88.4 |
| P-Tuning v1 | 79.1 ± 2.4 | 79.0 ± 1.1 | 75.9 ± 18.3 | 24.7 ± 17.6 | 71.0 |
| P-Tuning v2 | 94.0 ± 1.1 | 91.2 ± 0.9 | 89.4 ± 0.7 | 66.9 ± 1.5 | 87.8 |
| Kron. AoT P-Tuning (ours) | **94.2 ± 0.1** | 89.7 ± 0.9 | 89.3 ± 0.1 | 65.5 ± 1.9 | 87.5 |
| FC AoT P-Tuning (ours) | 94.1 ± 0.2 | **91.6 ± 0.8** | **89.6 ± 0.1** | 69.2 ± 0.9 | **88.8** |

Table 2: Results on the GLUE Dev set. Each result is median and std across several seeds, and the Macro column is a mean score across all tasks. We bolded the best results and underlined the second best results. Fine-tuning is omitted from comparison with other methods and was not bolded for visibility. See Section 4.2 for details.

$l = 24$), then storing all biases $P$ will exceed 1.2B parameters, while the model itself has roughly 350M parameters.

To overcome this limitation, we propose two reparametrizations of $P$ so that it can use fewer parameters during training.

The first is based on the Kronecker product (namely, **Kronecker AoT P-Tuning**). More specifically, we reparametrize $\boldsymbol{P}$ as

$$\boldsymbol{P} = (\boldsymbol{W}_L \otimes \boldsymbol{W}_M)\boldsymbol{W}_R, \tag{2}$$

where $\boldsymbol{W}_L \in \mathbb{R}^{a \times r}$, $\boldsymbol{W}_M \in \mathbb{R}^{b \times r}$, $\boldsymbol{W}_R \in \mathbb{R}^{r^2 \times d}$, $a$ and $b$ are selected in such a way so $a * b = |\boldsymbol{V}|$, $r$ is the factorization rank which is a hyperparameter to tune, and $\otimes$ denotes the Kronecker product.

With this reparametrization, training AoT P-Tuning becomes tractable. E.g., for RoBERTa-Large, with $a = 256$, $b = 200$, and $r = 20$, $\boldsymbol{P}$ will contain roughly 10M parameters, which is less than $3\%$ of the total number of parameters in the model[1].

The second approach to work with $\boldsymbol{P}$, which we used in our experiments, is based on passing the embeddings matrix $\boldsymbol{E}$ through a learnable Fully Connected network (namely, **FC AoT P-Tuning**). Thus, we reparametrize $\boldsymbol{P}$ as

$$\boldsymbol{P} = f(\boldsymbol{E}\boldsymbol{W}_1 + \boldsymbol{b}_1)\boldsymbol{W}_2 + \boldsymbol{b}_2, \tag{3}$$

where $\boldsymbol{W}_1 \in \mathbb{R}^{d \times r}$, $\boldsymbol{b}_1 \in \mathbb{R}^r$, $\boldsymbol{W}_2 \in \mathbb{R}^{r \times d}$, $\boldsymbol{b}_2 \in \mathbb{R}^d$, $f$ is a non-linearity, and $r$ is the mapping rank, which is also a hyperparameter to tune, same as for Kronecker AoT P-Tuning.

With FC AoT P-Tuning, we utilize knowledge stored in the pre-trained embeddings matrix $\boldsymbol{E}$, which should hypothetically perform better than training $\boldsymbol{P}$ from scratch as Kronecker AoT P-Tuning.

Note that for both Kronecker and FC AoT P-Tuning, we can evaluate only specific rows $\{\boldsymbol{P}_{x_i}, \ldots, \boldsymbol{P}_{x_n}\}$ for input sequence $\{x_1, \ldots, x_n\}$, making training more efficient.

For both reparametrizations, $\boldsymbol{P}$ could be fused once training is complete, and thus the rank of factorization $r$ does not affect inference speed. During the evaluation, there is no need to store the full $\boldsymbol{P}$ in GPU memory. Instead, it could be stored in RAM, and only rows of these matrices should be placed in GPU memory to be added to the hidden states before each layer.

From a certain perspective, choosing between AoT P-Tuning and P-Tuning is a trade-off between evaluation speed and RAM consumption during inference. If RAM is limited, then usual P-Tuning could be used at the cost of slower inference. In other cases, AoT P-Tuning is viable if there is enough RAM and inference speed is crucial. Although, in most cases, $\boldsymbol{P}$ matrices for different tasks could be easily stored in the RAM. For RoBERTa-Large, a single task parameter will require roughly 2.4Gb if stored in half-precision.

## 4 EXPERIMENTS

### 4.1 EXPERIMENTAL DETAILS

We compared AoT P-Tuning (Kronecker and FC reparametrizations of $\boldsymbol{P}$) with other fine-tuning methods capable of performing multi-task inference: P-Tuning v1, P-Tuning v2 on GLUE and SuperGLUE (Wang et al., 2018; 2019) Benchmarking Datasets. We also evaluated plain fine-tuning, LoRA, Adapters, and BitFit for reference. For each fine-tuning approach, we experimented with the RoBERTa-Base, RoBERTa-Large, and DeBERTa-XL backbone models.

For each task, we performed a grid hyperparameter search (see Appendix Table 5 for hyperparameter ranges). For RoBERTa models, we evaluated each hyperparameter set with 5 different seed values and reported median and std score values for each task. For DeBERTa-XL, we used to assess each hyperparameter assignment with a single seed due to longer training time. See Appendix Table 4 for a list of metrics used for each task.

---

[1]One may note that $256 * 200 = 51200 \neq 50265$. However, 50265 is difficult to factorize efficiently since $50265 = 1117 * 3^2 * 5$. Because of this, we chose to mostly factorize $\boldsymbol{P}$ in such a way as to make it slightly larger than the original vocabulary size. Doing so allows us to select more appropriate $a$ and $b$ from the perspective of parameter and computational efficiency.

| RoBERTa-Large | | | | |
|---|---|---|---|---|
| Model | RTE | COPA | WSC | WiC |
| Fine-Tuning | 88.1 ± 1.5 | 87.0 ± 10.2 | 80.8 ± 6.3 | 73.8 ± 1.6 |
| Adapters | 87.7 ± 18.5 | 89.0 ± 10.1 | 77.9 ± 9.8 | **73.5 ± 1.0** |
| LoRA | 87.4 ± 1.6 | **91.0 ± 8.5** | **79.8 ± 10.6** | 71.9 ± 1.2 |
| BitFit | 87.7 ± 0.8 | **91.0 ± 2.3** | 71.2 ± 6.7 | 71.3 ± 9.5 |
| P-Tuning v1 | 62.8 ± 2.3 | 75.0 ± 4.3 | 66.3 ± 1.3 | 64.1 ± 0.9 |
| P-Tuning v2 | 87.4 ± 1.5 | 87.0 ± 6.3 | 75.0 ± 7.7 | 70.8 ± 1.5 |
| Kron. AoT P-Tuning (ours) | 84.8 ± 1.3 | 72.0 ± 9.1 | 67.3 ± 3.0 | 71.0 ± 1.0 |
| FC AoT P-Tuning (ours) | **88.4 ± 0.9** | 85.0 ± 10.1 | **79.8 ± 4.1** | 72.1 ± 1.5 |
| | MultiRC | CB | BoolQ | Macro |
| Fine-Tuning | 83.3 ± 1.1 | 97.3 ± 2.8 | 85.6 ± 0.3 | 85.1 |
| Adapters | **83.7 ± 20.3** | **100.0 ± 0.0** | **85.7 ± 10.6** | **85.4** |
| LoRA | 75.7 ± 17.4 | **100.0 ± 2.6** | 84.6 ± 0.6 | 84.3 |
| BitFit | 82.5 ± 0.6 | **100.0 ± 0.7** | 85.4 ± 1.0 | 84.2 |
| P-Tuning v1 | 54.3 ± 2.9 | 81.4 ± 3.0 | 64.3 ± 1.2 | 66.9 |
| P-Tuning v2 | 82.4 ± 0.6 | **100.0 ± 0.8** | 85.0 ± 0.6 | 83.9 |
| Kron. AoT P-Tuning (ours) | 82.8 ± 0.8 | 97.3 ± 2.3 | 84.8 ± 0.5 | 80.0 |
| FC AoT P-Tuning (ours) | 82.7 ± 19.3 | **100.0 ± 0.0** | 85.5 ± 10.3 | 84.8 |
| DeBERTa-XL | | | | |
| Model | RTE | COPA | WSC | WiC |
| Fine-Tuning | 89.9 | 96.0 | 76.9 | 75.9 |
| Adapters | 90.3 | 96.0 | 89.4 | **77.3** |
| LoRA | 90.3 | 97.0 | 89.4 | 75.5 |
| BitFit | 89.2 | 97.0 | 86.5 | 73.7 |
| P-Tuning v1 | 78.3 | 90.0 | 67.3 | 66.8 |
| P-Tuning v2 | 90.6 | 97.0 | 89.4 | 76.5 |
| Kron. AoT P-Tuning (ours) | 88.8 | 96.0 | 87.5 | 71.8 |
| FC AoT P-Tuning (ours) | **91.0** | **98.0** | **94.2** | 74.1 |
| | MultiRC | CB | BoolQ | Macro |
| Fine-Tuning | 84.3 | 98.4 | 86.7 | 86.9 |
| Adapters | 86.7 | 97.3 | **88.9** | 89.4 |
| LoRA | 86.0 | **100.0** | 88.3 | **89.5** |
| BitFit | 85.2 | **100.0** | 86.5 | 88.3 |
| P-Tuning v1 | 82.1 | 93.8 | 79.4 | 79.7 |
| P-Tuning v2 | **87.1** | 97.3 | 87.0 | 89.3 |
| Kron. AoT P-Tuning (ours) | 86.3 | 83.1 | 87.3 | 85.8 |
| FC AoT P-Tuning (ours) | 86.5 | 92.3 | 88.1 | 89.2 |

Table 3: Results on the SuperGLUE Dev set. For RoBERTa-Large, each result is median and std across several seeds, and the Macro column is a mean score across all tasks. For DeBERTa-XL, we evaluated each hyperparameter assignment with a single seed and reported its metric score. We bolded the best results and underlined the second best results. Fine-tuning is omitted from comparison with other methods and was not bolded for visibility. See Section 4.2 for details.

We used the Adam (Kingma & Ba, 2015) optimizer with a constant learning rate for each task. We stopped training once the validation metric stopped increasing (see the "patience" parameter in Appendix Table 6).

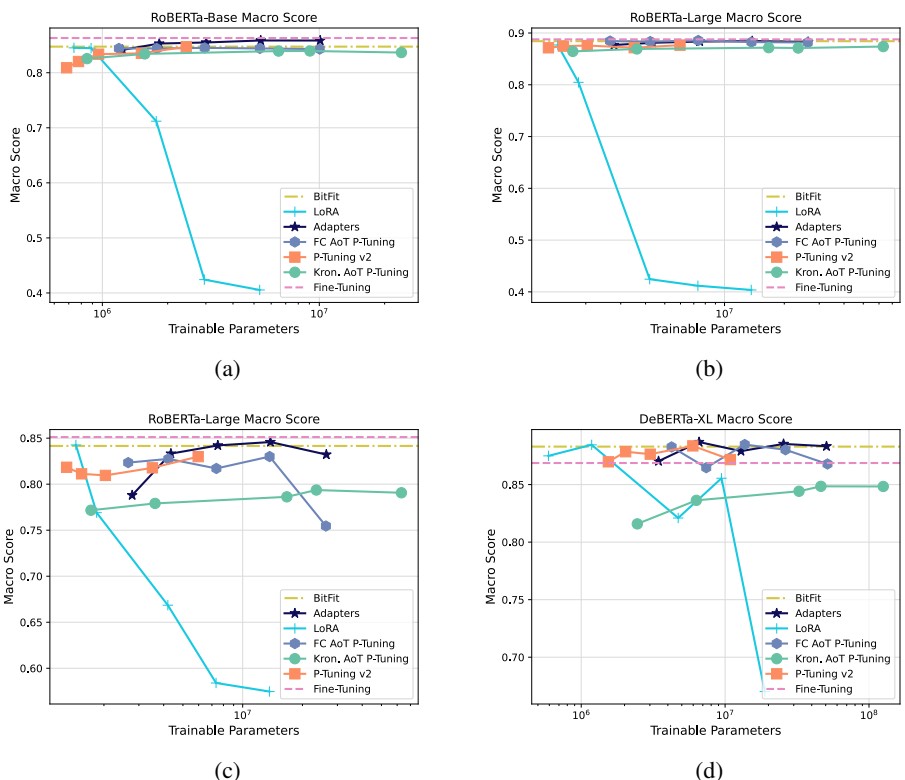

Figure 3: (a-b) GLUE macro scores for AoT P-Tuning, P-Tuning v1, and P-Tuning v2 with RoBERTa-Base and RoBERTa-Large models. (c-d) SuperGLUE macro score for RoBERTa-Base and DeBERTa-XL models. P-Tuning v2 performing on par with or worse than AoT P-Tuning across different prefix sizes. See Section 4.2 for details.

For Kronecker AoT P-Tuning with RoBERTa models, we parametrized the matrix $\boldsymbol{P} = (\boldsymbol{W}_L \otimes \boldsymbol{W}_M)\boldsymbol{W}_R$ with $a = 256$, and $b = 200$, while for DeBERTa, we used $a = b = 360$. $\boldsymbol{W}_L$ and $\boldsymbol{W}_M$ were initialized randomly, while $\boldsymbol{W}_R$ was initialized as a zero matrix. For FC AoT P-Tuning, we initialized $\boldsymbol{W}_1$ randomly, while $\boldsymbol{W}_2$, $\boldsymbol{b}_1$, and $\boldsymbol{b}_2$ were initialized with zeros. For Kronecker AoT P-Tuning, we applied dropout (Srivastava et al., 2014) to the $\boldsymbol{P}_x$ with a fixed probability equal to $0.1$. In contrast, for FC AoT P-Tuning, we applied dropout to $\boldsymbol{E}$ before multiplying it with $\boldsymbol{W}_1$.

Each experiment was run on a single NVIDIA A100 GPU with a total computation time of roughly 1200 days.

## 4.2 RESULTS

See Tables 2, 3 for the results of trained models. We observed that FC AoT P-Tuning performed better than Kronecker AoT P-Tuning, and hypothesize that this result is mostly caused by the fact that FC reparametrization utilized a pre-trained embedding matrix rather than learning biases from scratch.

For RoBERTa-Base, FC AoT P-Tuning performed on par with P-Tuning v2 and produced the same Macro score. For RoBERTa-Large, FC AoT P-Tuning outperformed P-Tuning v2 on GLUE tasks and showed a Macro score equal to plain Fine-Tuning. AoT P-Tuning with DeBERTa-XL performed on par with P-Tuning v2 (89.2 vs 89.3 macro scores respectively).

We also observed that both AoT P-Tuning reparametrizations mainly showed a lower variance of metrics across different seeds. Note that P-Tuning v1 showed unstable performance and improved

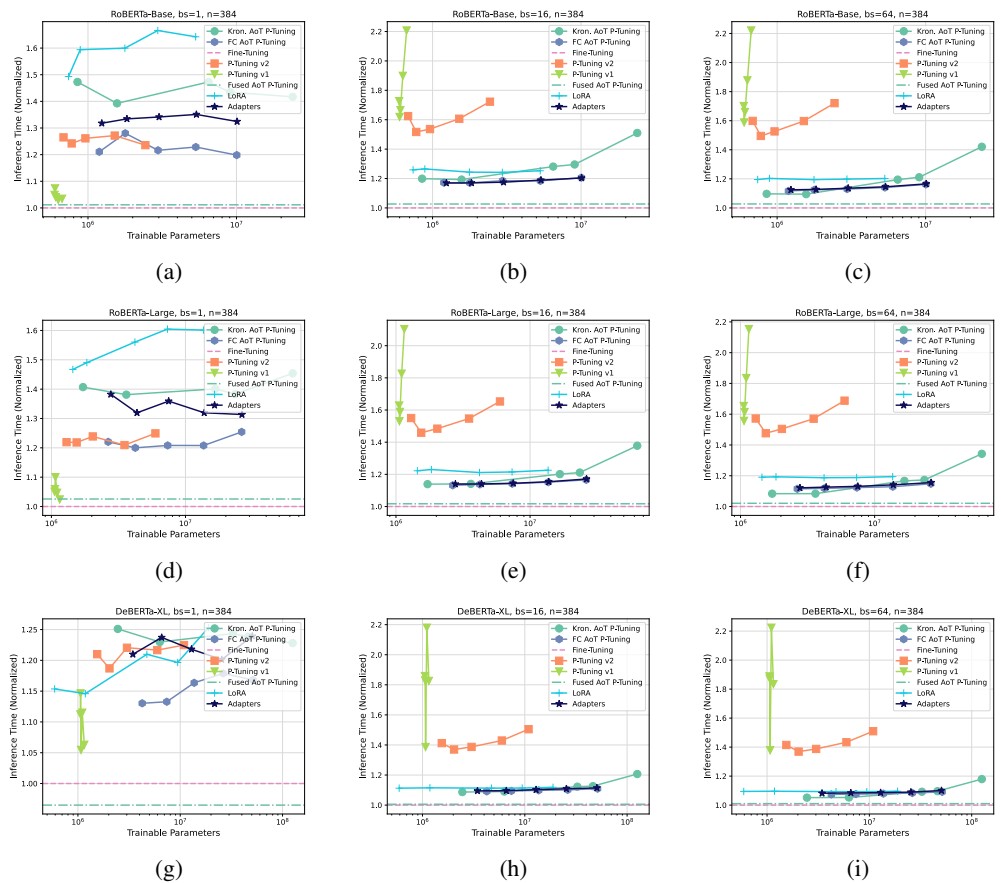

Figure 4: Speed measurements for baseline methods with sequence length equal to $384$ for different back-bone models. See Appendix Figure 9 for results with other sequence lengths and Section 4.3 for details.

results with RoBERTa-Base (although still underperforming by a large margin when compared to other methods).

See Figure 3 for macro scores of P-Tuning v2 and AoT P-Tuning with different prefix lengths $p$ and prefix ranks $r^2$. We observed that P-Tuning v2 performed worse for RoBERTa-Base with shorter prompt lengths and was comparable to or better than AoT P-Tuning when $p > 50$. For GLUE tasks with RoBERTa-Large, FC AoT P-Tuning performed better for all prefixes $p$, while dropping performance for large rank $r$. For DeBERTa-XL, both P-Tuning v2 and FC AoT P-Tuning performed on par. We also provide per-task results with different prefix scales (see Appendix Figures 5, 7). It is notable that in most cases, P-Tuning v2 suffers from a small prefix size $p$ for Base and Large models, and achieves results comparable with AoT P-Tuning with a larger $p$ (which corresponds with the results in Figure 3). At the same time, FC AoT P-Tuning mostly showed stable performance across different ranks $r$, only performing unstably on a MultiRC task with a large rank $r$.

Compared with LoRA and Adapters methods, RoBERTa-Large Adapters showed a higher Macro score with SuperGLUE than all P-Tuning methods, including AoT P-Tuning, even while showing larger variance across task scores. For DeBERTa-XL, LoRA and Adapters marginally outperformed P-Tuning methods on the macro score. BitFit performed worse than FC AoT P-Tuning, except for RoBERTa-Base back-bone where it performed the same. From such a perspective, if the ability to

---

[2]Note that the best macro result across different scales of prefixes in these Figures differs from the macro result from Tables 2 and 3, since the macro score from Tables 2 and 3 aggregates scores with different prefix scales.

perform multi-task inference without computational overhead is crucial, then AoT P-Tuning will not dramatically reduce performance compared to LoRA and Adapters and could be used. Note that the gap in performance between AoT P-Tuning and Adapters and LoRA became marginal with model growth.

With per-task Expected Validation Performance (EVP) (Dodge et al., 2019), we observed that AoT P-Tuning highly depends on the number of hyperparameter assignments (see Appendix Figures 6, 8). Although, in most cases, using less than 100 hyperparameter assignments for AoT P-Tuning is enough for it to outperform P-Tuning v2, which is not crucial in most cases.

We also analyzed trained $P$ matrices for FC AoT P-Tuning with the DeBERTa-XL model. See Appendix Section C for more details.

### 4.3 INFERENCE SPEED OVERHEAD

In Figure 4 and Appendix Figure 9, we investigated the computational overhead of AoT P-Tuning compared to other baselines.

We estimated inference time for RoBERTa-Base, RoBERTa-Large, and DeBERTa-XL models with batch sizes $\in [1, 16, 64]$ and sequence lengths $\in [64, 128, 384]$. For batch size equal to 1, we evaluated the model 300 times, and 100 times for other values. We report mean values of inference time normalized by the inference time of the vanilla model (i.e., plain fine-tuning).

We evaluated AoT P-Tuning on two setups. For the first setup, we fused $P$ so that the model can perform on its top speed. We did not perform fusing for the second setup. While a lack of fusing is not required to use AoT P-Tuning, we report these results for reference. We also report LoRA results for the experiments in which we did not fuse weights. This setup makes it possible to use LoRA in a multi-task setup (see Section 3.1 for details).

We observed that the proposed method performed differently depending on the experiment's parameters. AoT performed with computational overhead for a small model (i.e., RoBERTa-Base), small sequence length, and small batch size. We observed $12\%$ overhead on inference speed compared to plain fine-tuning for RoBERTa-Base with batch size 1 and sequence length 64. However, AoT P-Tuning still performed faster than other baselines by a large margin (i.e., LoRA for RoBERTa-Base with batch size 1 and sequence length 384 added $50 - 70\%$ computational overhead compared to fine-tuning).

Once model size or input size is increased, we observed that the overhead of adding biases for AoT P-Tuning becomes negligible. The proposed method performed the same as plain fine-tuning (for some experiments, we observed $1 - 2\%$ overhead, while for others, AoT P-Tuning performed even faster than fine-tuning, which is a variation in inference time measurements). For the most practical setup with large models (DeBERTa-XL), small batch size ($= 1$) and long sequence length ($= 384$), AoT P-Tuning performed slightly faster than fine-tuning, while other methods performed $12 - 25\%$ slower.

## 5 CONCLUSION AND FUTURE WORK

In this paper, we proposed AoT P-Tuning, which is a new method for parameter-efficient fine-tuning of pre-trained models, and two reparametrizations of learnable weights for this method.

We observed that AoT P-Tuning performed on par or better than P-Tuning v2 based on the macro scores of GLUE and SuperGLUE Benchmarking Datasets. In addition, compared to the LoRA and Adapters methods with a large backbone model (i.e., DeBERTa-XL), we observed a speed increase at the cost of only a marginal performance drop.

We experimented with two reparametrizations based on the Kronecker product and FC network. It is possible to explore other possible reparametrizations for weight $P$, which could further increase the performance of the proposed method. In addition, while we proposed a simple method, there are many possible architectural changes which could also boost the performance of AoT P-Tuning and reduce the number of necessary hyperparameter assignments.

## 6 REPRODUCIBILITY

We organized our experiments in such a way that makes it possible to quickly reproduce all the results reported in our paper.

`tools/run_sweeps.py` runs wandb sweeps used in our hyperparameter search and outputs wandb agent commands to run these sweeps. Once the hyperparameter search is finished, `tools/glue_results.py` and `tools/superglue_results.py` scripts are used to parse results from these sweeps and output plots reported by us.

The data processing pipeline for all datasets used in this paper can be found in the `zarya/data_processing.py` script. It could be helpful for tasks such as WSC, where several ways to tokenize input sequences could affect the resulting performance.

We also provided `requirements.txt` (with fixed dependencies versions) and a Dockerfile with our source files to make it possible to quickly reproduce our training setup without issues with versions of libraries used for our experiments.

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

## A  BACKGROUND

### A.1  EVALUATION OF TRANSFORMER

Having an input sequence $\boldsymbol{x} = \{x_1, \ldots, x_n\}$, where $x_i$ is token index, the embeddings of input texts are evaluated as $\boldsymbol{H}^0 = \{\boldsymbol{E}_{x_1}, \ldots, \boldsymbol{E}_{x_n}\}$, where $\boldsymbol{E} \in \mathbb{R}^{|\boldsymbol{V}| \times d}$ is the embeddings matrix, $|\boldsymbol{V}|$ is the vocabulary size, $d$ is the size of the hidden state of the model, and $\boldsymbol{E}_{x_i}$ is an embedding of the token $x_i$. Hidden states $\boldsymbol{H}^i$ are then passed to the $(i+1)$-th layer of the Transformer to evaluate $\boldsymbol{H}^{i+1}$ with a total $l$ number of layers. To do so, $\boldsymbol{H}^i$ are first mapped through three matrices $\boldsymbol{W}_Q$, $\boldsymbol{W}_K$, $\boldsymbol{W}_V \in \mathbb{R}^{d \times d}$ to get $\boldsymbol{Q}$, $\boldsymbol{K}$ and $\boldsymbol{V}$, which are then used to evaluate the attention layer's results as:

$$\boldsymbol{A} = \text{attention}(\boldsymbol{Q}, \boldsymbol{K}, \boldsymbol{V}) = \text{softmax}(\frac{\boldsymbol{Q}\boldsymbol{K}^T}{\sqrt{d}})\boldsymbol{V} \in \mathbb{R}^{n \times d}. \tag{4}$$

After $\boldsymbol{A}$ is evaluated, it is passed through the remaining layers[3], including residual connections and FC layers to get $\boldsymbol{H}^{i+1}$. Here and later, we omit the layer index $i$ for attention result $\boldsymbol{A}$ for visibility.

### A.2  P-TUNING V1

Having a pre-trained Transformer LM with parameters $\Theta$, instead of fine-tuning all parameters of this model on a downstream task, it is possible to define soft prompts $\boldsymbol{P} \in \mathbb{R}^{p \times d}$(Liu et al., 2021b), where $p$ is the length of prompt. $\boldsymbol{P}$ is then concatenated to input sequence embeddings as:

$$\boldsymbol{H}'^0 = \text{concat}(\boldsymbol{P}, \boldsymbol{H}^0) \in \mathbb{R}^{(p+n) \times d}. \tag{5}$$

Then, only $\boldsymbol{P}$ and Classification Head are fine-tuned on a downstream task, while $\Theta$ remains frozen[4]. Such parametrization of fine-tuning makes it possible to perform multi-task inference.

### A.3  P-TUNING V2

Instead of concatenation of a single prompt $\boldsymbol{P}$ to the $\boldsymbol{H}^0$, Liu et al. (2021a) proposed to concatenate soft prefixes at each layer of the Transformer model. To apply P-Tuning v2, soft prefixes $\boldsymbol{P}_K, \boldsymbol{P}_V \in \mathbb{R}^{p \times d}$ are defined for each layer and concatenated to the $\boldsymbol{K}$ and $\boldsymbol{V}$ matrices before evaluating the attention $\boldsymbol{K}' = \text{concat}(\boldsymbol{P}_K, \boldsymbol{K})$, $\boldsymbol{V}' = \text{concat}(\boldsymbol{P}_V, \boldsymbol{V})$. Then, Attention is evaluated as follows:

$$\boldsymbol{A}' = \text{attention}(\boldsymbol{Q}, \boldsymbol{K}', \boldsymbol{V}'), \tag{6}$$

where $i$-th component of $\boldsymbol{A}'$ could be then written as:

$$\boldsymbol{A}'_i = \sum_{j=1}^{p} \boldsymbol{a}_j(\boldsymbol{Q}_i, \boldsymbol{K}')\boldsymbol{P}_{V_j} + \sum_{k=1}^{n} \boldsymbol{a}_{k+p}(\boldsymbol{Q}_i, \boldsymbol{K}')\boldsymbol{V}_k. \tag{7}$$

---

[3]In fact, Transformer architecture implies evaluation of multi-head Attention. We omit this in this paper for simplicity since all derivations could be easily extended on the multi-head case.

[4]Original implementation of P-Tuning v1 (Liu et al., 2021b) implied utilizing the LM Head of a pre-trained model instead of training a Classification Head. However, Liu et al. (2021a) later showed that using a separate Classification Head performs marginally better.

Note that $\boldsymbol{a} \in \mathbb{R}^{p+n}$ are attention weights for the $i$-th token (we omit the $i$-th index for simplicity) and thus $\sum_{j=1}^{p+n} \boldsymbol{a}_j = 1$.

As for P-Tuning v1, only parameters of soft prefixes $\boldsymbol{P}_K, \boldsymbol{P}_V$ and Classification Head are optimized on a downstream task while freezing the parameters of a backbone model.

## B  INTUITION BEHIND AoT P-TUNING AND CONNECTION TO THE P-TUNING

Having $\boldsymbol{H}'$, after passing through $\boldsymbol{W}_Q$, $\boldsymbol{W}_K$, and $\boldsymbol{W}_V$ we obtain $\boldsymbol{Q}'$, $\boldsymbol{K}'$, and $\boldsymbol{V}'$. Note that $\boldsymbol{V}' = \boldsymbol{H}\boldsymbol{W}_V + \{\boldsymbol{P}_{x_1}, \ldots, \boldsymbol{P}_{x_n}\}\boldsymbol{W}_V \overset{\text{def}}{=} \boldsymbol{V} + \boldsymbol{P}_x\boldsymbol{W}_V$.

The result of evaluating Attention with AoT P-Tuning could be seen as:

$$\boldsymbol{A}'_i = \sum_{j=1}^{n} \boldsymbol{a}_j(\boldsymbol{Q}'_i, \boldsymbol{K}')\boldsymbol{P}_{x_j}\boldsymbol{W}_V + \sum_{j=1}^{n} \boldsymbol{a}_j(\boldsymbol{Q}'_i, \boldsymbol{K}')\boldsymbol{V}_j. \tag{8}$$

From such a perspective, there is a clear connection between AoT P-Tuning (Equation 8) and P-Tuning v2 (Appendix Equation 7) with the following changes:

1. For AoT P-Tuning, attention weights $\boldsymbol{a}_j$, $j \in \overline{1, l}$ are used for both terms in Equation 8.

2. For AoT P-Tuning, attention is evaluated on modified $\boldsymbol{Q}'$. In addition, there is a difference in the form of dependency of $\boldsymbol{K}'$ and $\boldsymbol{V}'$ on prefix weight. For AoT P-Tuning, we add prefixes to $\boldsymbol{K}$ and $\boldsymbol{V}$, while for P-Tuning v2, prefixes are concatenated to these matrices.

3. For AoT P-Tuning, the first term of Equation 8 implies evaluation of Attention with a prompt which is dependent on the input text, while for P-Tuning v2, the prompt $\boldsymbol{P}_V$ is constant.

Considering Equation 8, AoT can be seen as a form of the P-Tuning method, for which we embed prefixes before evaluating the attention layer[5].

## C  ANALYSIS OF TRAINED WEIGHTS

We investigated trained $\boldsymbol{P}$ matrices for WSC, COPA, CB, and RTE tasks with the DeBERTa-XL model. Since FC AoT P-Tuning performed better than Kronecker factorization, we selected this reparametrization method to report the results.

More specifically, we sorted rows of $\boldsymbol{P}$ matrices for each layer measured by the $L_2$ norm and reported the appropriate tokens for these rows. See Tables 7, 8, 10, 9 for results.

For the WSC task, there is a clear interpretation of trained rows for $\boldsymbol{P}$, since rows with a large $L_2$ norm represent tokens responsible for pronouns and names, which is crucial for solving WSC. For the COPA task, we observed that the model tends to assign large norms for verb tokens. For the RTE and CB tasks, $\boldsymbol{P}$ also assigns large norms for name tokens, which often occur in the training data, while CB primarily modifies adverbs for later layers.

---

[5]It is possible to think of AoT P-Tuning as a method which adds bias **after** the evaluation of the Transformer layer. In this case, it could be seen as a method that directly models the result of the evaluation of P-Tuning v2 with a slightly different computation order. However, we believe that this way is more difficult to consider.

| Task | Metric | Task | Metric |
|------|--------|------|--------|
| CoLA | Mattews Correlation | BoolQ | Accuracy |
| MRPC | $\frac{\text{Accuracy}+\text{F1}}{2}$ | CB | $\frac{\text{Accuracy}+\text{F1}}{2}$ |
| RTE | Accuracy | RTE | Accuracy |
| SST-2 | Accuracy | COPA | Accuracy |
| MNLI | Accuracy | MultiRC | $\frac{\text{Accuracy}+\text{F1}}{2}$ |
| QNLI | Accuracy | WSC | Accuracy |
| QQP | $\frac{\text{Accuracy}+\text{F1}}{2}$ | WiC | Accuracy |
| STSB | $\frac{\text{Pearson}+\text{Spearman}}{2}$ | | |

Table 4: Metrics used in our experiments for each task. See Section 4.1 for more details.

| Parameter | Range |
|-----------|-------|
| All Tasks, except RTE | |
| P-Tuning v1/v2/AoT | |
| batch size | 16, 64 |
| learning rate | 1e−4, 5e−4, 5e−3, 1e−3 |
| $p$ | 5, 10, 20, 50, 100 |
| LoRA $r$ | 2, 4, 16, 32, 64 |
| Adapters $r$ | 16, 32, 64, 128, 256 |
| Kron. $r$ | 5, 10, 25, 30, 50 |
| FC $r$ | 32, 64, 128, 256, 512 |
| Fine-Tuning | |
| learning rate | 1e−5, 5e−5, 1e−4, 5e−4, 5e−3 |
| RTE | |
| batch size | 16, 32, 64, 128 |
| learning rate | 1e−5, 5e−5, 1e−4, 5e−4, 5e−3, 1e−3, 2e−3, 1e−2 |

| Parameter | Range |
|-----------|-------|
| P-Tuning v1/v2/AoT | |
| batch size | 16, 32, 64 |
| learning rate | 5e−5, 1e−4, 3e−4, 5e−4, 1e−3, 2e−3, 5e−3 |
| $p$ | 5, 10, 20, 50, 100 |
| LoRA $r$ | 2, 4, 16, 32, 64 |
| Adapters $r$ | 16, 32, 64, 128, 256 |
| Kron. $r$ | 5, 10, 25, 30, 50 |
| FC $r$ | 32, 64, 128, 256, 512 |
| Fine-Tuning | |
| learning rate | 1e−5, 5e−5, 1e−4, 5e−4, 5e−3 |

Table 5: Hyperparameter ranges used in experiments with GLUE and SuperGLUE benchmarking datasets for RoBERTa (left) and DeBERTa (right) models. $p$ is the prompt length used for P-Tuning v1/v2, and $r$ is the rank of weight factorization used for AoT P-Tuning (See Section 3.3). For GLUE experiments, each hyperparameter set was evaluated with different seed values. See Section 4.1 for more details.

| | RTE | MNLI, QQP | QNLI | Other Tasks | WiC | CB, COPA, WSC | MultiRC | Other Tasks |
|---|---|---|---|---|---|---|---|---|
| Epochs | 200 | 5 | 10 | 100 | 500 | 500 | 10 | 100 |
| Patience | 20 | 2 | 2 | 10 | 20 | 100 | 4 | 10 |

Table 6: The number of maximum epochs used for each GLUE and SuperGLUE Task. Once the Dev score stopped increasing for "patience" steps, training was halted. See Section 4.1 for more details.

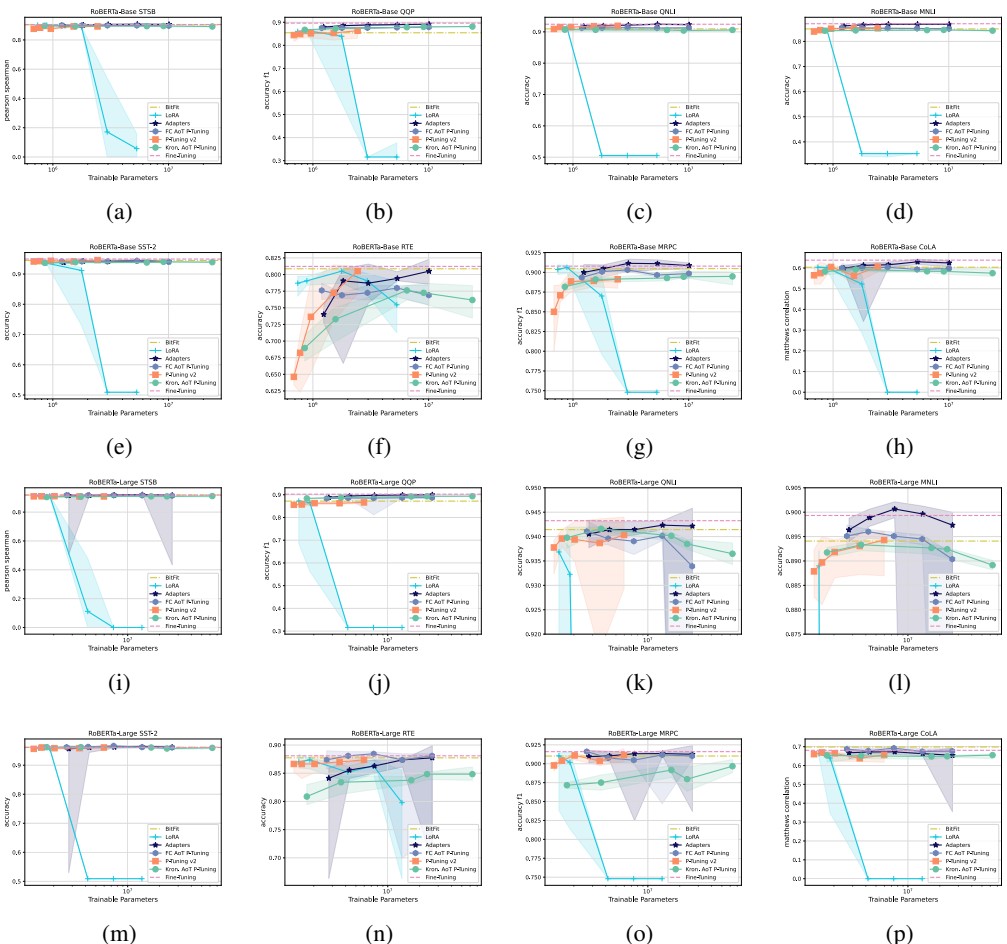

Figure 5: Per-task GLUE Benchmarking Dataset results for a different number of trained parameters of P-Tuning v2 and AoT P-Tuning with RoBERTa-Base (a-h) and RoBERTa-Large (i-p). We also provide results of plain fine-tuning for reference. See Section 4.2 for more details.

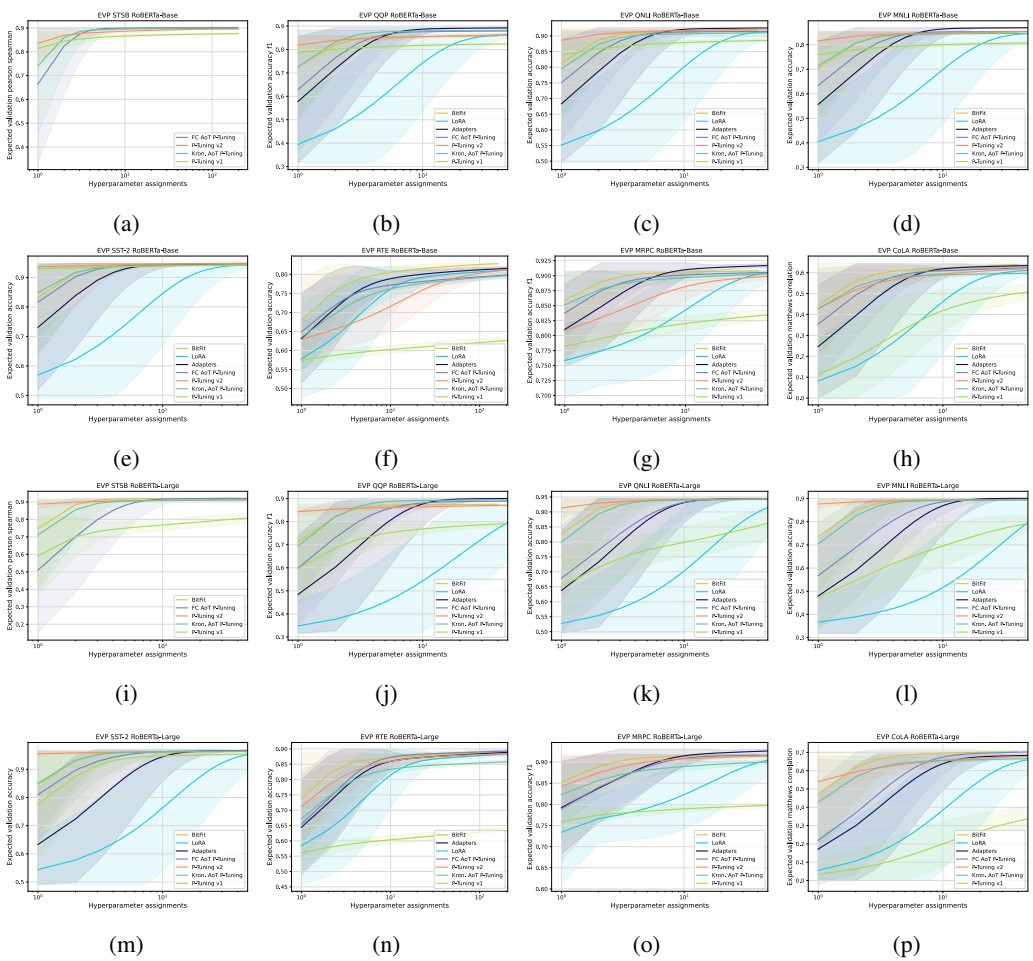

Figure 6: Expected Validation Performance (Dodge et al., 2019) of trained models with GLUE Benchmarking Datasets for RoBERTa-Base (a-h) and RoBERTa-Large (i-p). See Section 4.2 for more details.

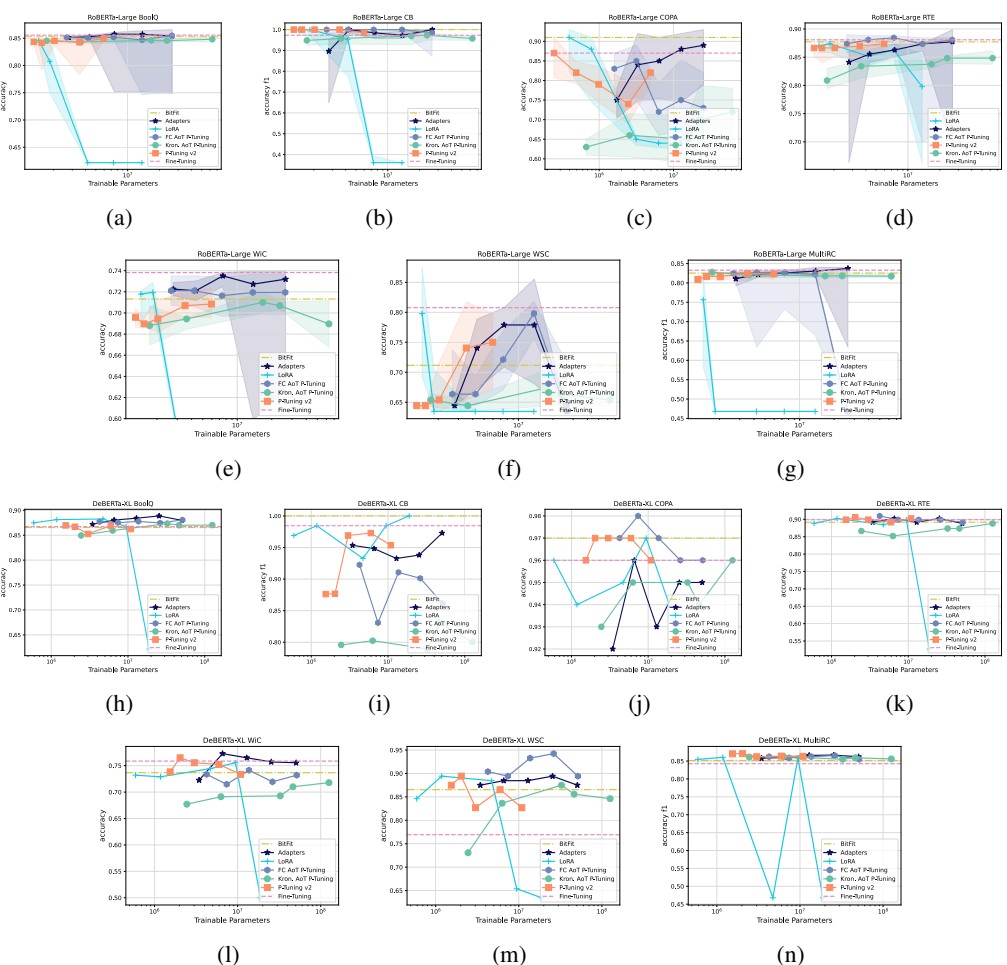

Figure 7: Per-task SuperGLUE Benchmarking Dataset results for a different number of trained parameters of P-Tuning v2 and AoT P-Tuning with RoBERTa-Large (a-g) and RoBERTa-Large (h-n). We also provide results of plain fine-tuning for reference. See Section 4.2 for more details.

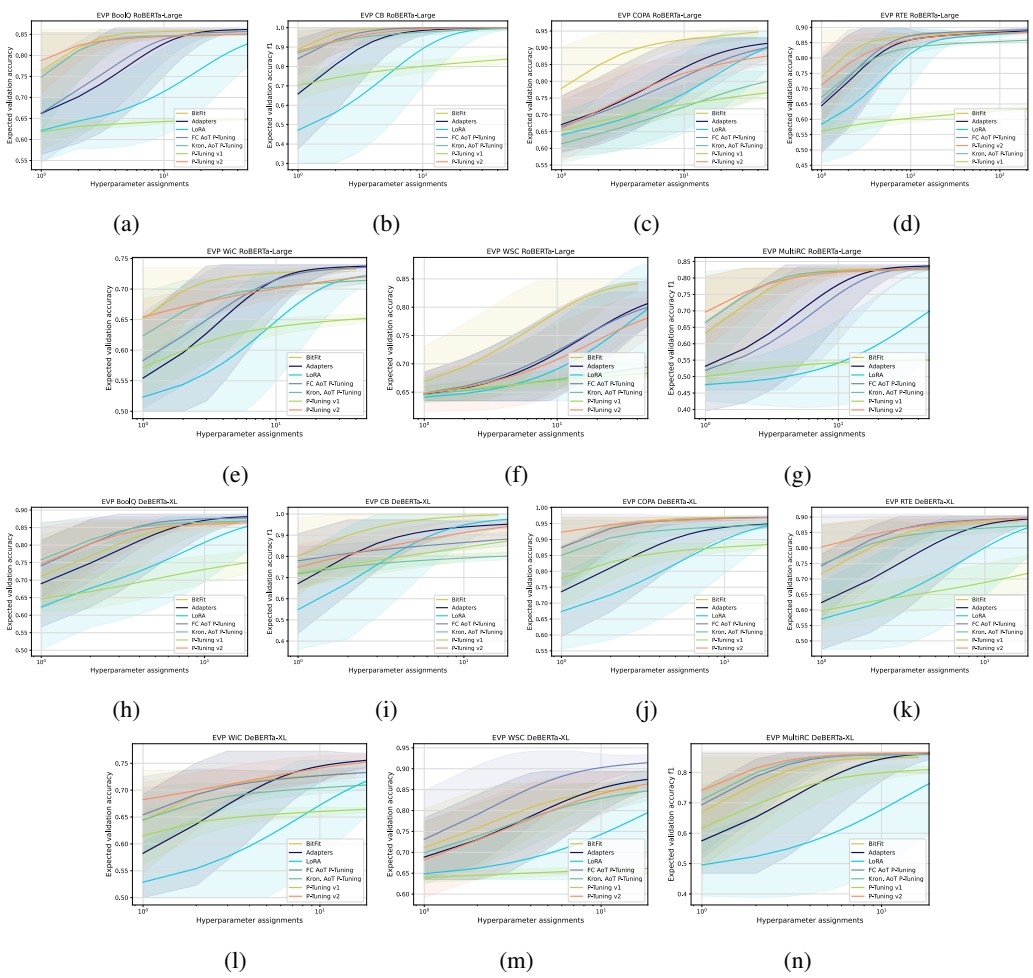

Figure 8: Expected Validation Performance (Dodge et al., 2019) of trained models with SuperGLUE Benchmarking Datasets for RoBERTa-Base (a-g) and RoBERTa-Large (h-n). See Section 4.2 for more details.

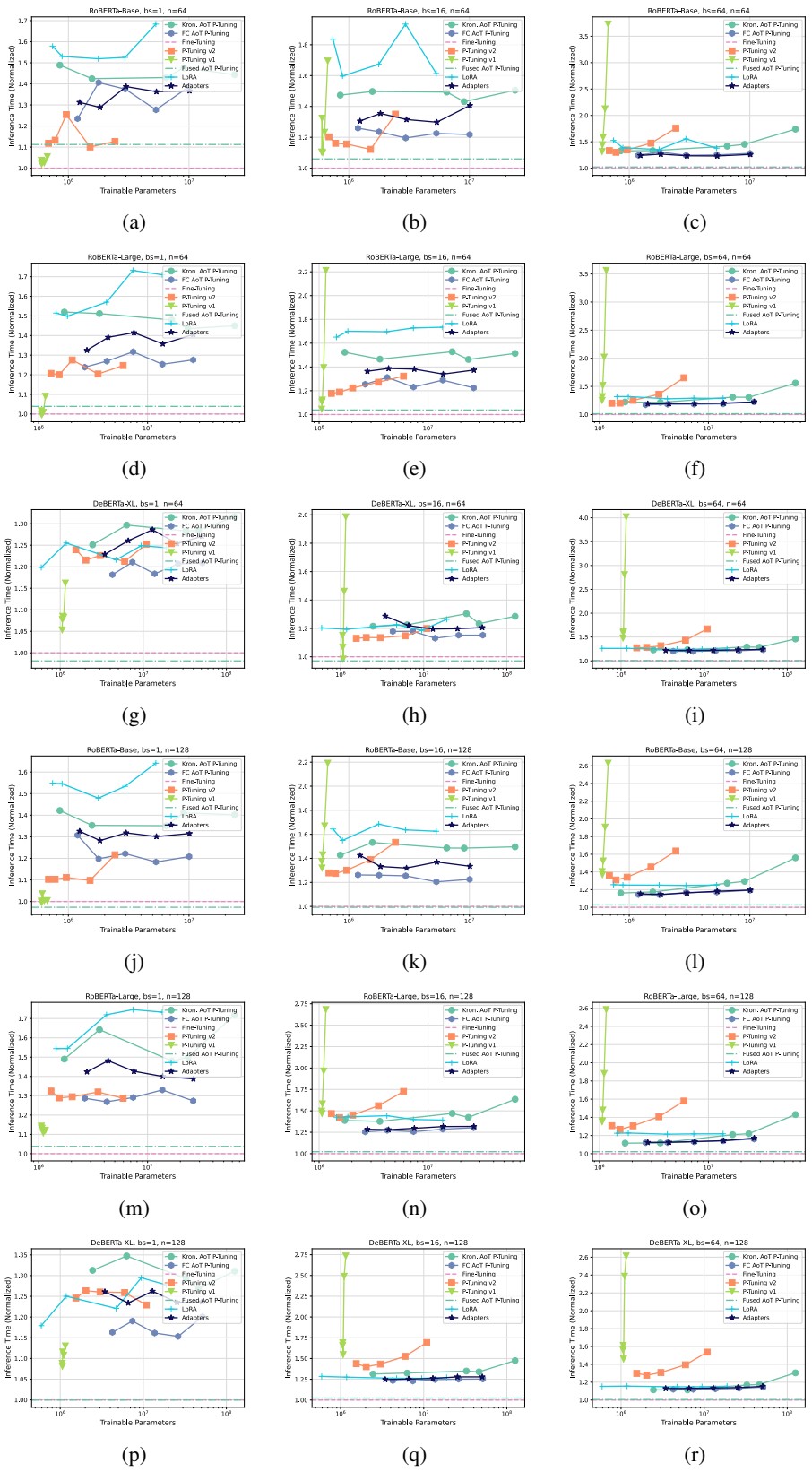

Figure 9: Speed measurement for baseline methods with sequence length $\in [16, 64]$ for different backbone models. See Section 4.3 for details.

| l# | Tokens $x$ with largest norm $\|\boldsymbol{P}_x\|_2$ |
|---|---|
| 0 | likes, a, is, loves, was, to, as, wants, s, ., pony, eded, himself, Man, were, and, I, has, I, are, Frank, ., hates, As, A, A, like, It, crop, Frank, After, ,, joins, As, Eric, Likes, It, just, would, onna, him, To, behaving, after, in, because, behaves, Is, We, Like |
| 5 | ,, ., narrower, doorway, backdoor, window, lousy, shortest, nicer, checkpoint, knob, thinner, narrowing, oub, quieter, BAD, ;, VID, rectangle, tighter, crappy, intruder, tongues, fing, rimination, blocker, and, raiding, detector, unmarked, sharper, knife, coolest, thicker, hoops, DOWN, lightsaber, asshole, millisec, KEY, sharp, token, slashing, Defenders, jug, Donna, slider, wedge, dding, kb |
| 10 | her, Her, herself, him, above, she, out, hers, him, HER, She, she, care, HER, above, Her, bold, CARE, cared, over, harder, louder, Above, smarter, sooner, her, cares, better, Out, vind, stronger, She, taller, tougher, Him, ahead, so, HIM, Susan, happier, up, Harry, aloud, higher, Above, SHE, could, apart, barking, inem |
| 22 | ., there, dry, for, There, Her, her, sword, the, arse, wy, dry, duc, The, it, took, cr, Rig, og, There, landing, the, wide, centrally, red, grass, sw, oa, above, engine, FT, spir, cd, Coun, Ross, there, ws, guy, starter, mans, aniel, green, freely, d, wide, stall, far, artz, THERE, didn |
| 32 | it, me, olit, Polit, Pat, him, Private, Susan, pat, he, her, Self, Ins, Doc, Coun, Ang, Aut, Sil, ochond, me, Nob, IT, Senator, Professional, Dri, itized, Je, Capt, Hillary, Whe, He, Kid, Registered, itious, Michelle, Political, It, Shut, Phot, BIT, Politics, Bit, Jacob, ruct, Young, HE, Tu, them, Mot, itu |
| 37 | him, they, it, they, her, them, their, his, he, it, its, hers, theirs, was, he, Susan, old, older, THEY, They, ITS, forth, Georg, Thom, Tom, erved, Carl, Anna, nob, anos, itans, to, Eric, itcher, Harry, Tim, Jen, them, Kid, Jeremy, JOHN, Jennifer, hands, Todd, put, Thomas, she, Dan, Michelle, s |
| 46 | chased, erved, house, houses, life, chasing, market, self, chase, raised, chester, hunt, castle, HOU, atics, Singer, western, ogenous, rounded, stretched, esian, essed, omorphic, horse, SER, central, ledge, hole, asio, Self, Self, iverse, oker, Judd, DF, aday, paced, ourced, erness, Barkley, scape, sey, ationally, owned, landers, ded, study, directed, OWN, produced |

Table 7: Tokens with the largest $L_2$ norm of $\boldsymbol{P}$ entries for the WSC task. See Section C for more details.

| $l$# | Tokens $x$ with largest norm $\|\boldsymbol{P}_x\|_2$ |
|---|---|
| 0 | fit, Loud, as, Air, Upon, Sets, Bound, Apart, scratched, sets, fit, Upon, hosted, Shot, Unt, Host, fitt, Sight, atri, Ocean, ceed, ashore, set, enture, underwent, planes, boats, Waves, Ali, shi, Active, Set, Atmosp, Airways, Host, chat, Endless, pelled, rew, ached, unct, fitted, Proud, flu, itable, anson, Bound, Assets, host, sets |
| 5 | set, Set, sets, Set, SET, Setting, setting, SET, set, padd, Setting, Sets, sets, bed, Cause, setting, the, cause, tread, itch, paddle, cause, thirsty, Khe, he, anned, this, ?, of, Cause, What, bidding, This, This, what, What, a, his, lic, The, wish, fugitive, they, Bed, Air, wake, conscience, ., crowd, Let |
| 10 | ?, ?, ?!, ??, ., ?", "?, )?, ???, '?, set, .?, !?, ?), !, ????, ...?, set, ?', ,, Set, Set, ed, to, ??, ????????, as, ?????, ?)., setting, ?,, ?'", ?], sets, ..., -, lt, —, :, lic, ???, led, ur, . . . , punching, of, t, ?"., sets, um |
| 22 | What, set, What, out, Set, sets, on, to, '?, what, Set, in, WHAT, Sets, Setting, WHAT, from, Setting, dropped, of, Dig, Got, ?', set, Exper, Gets, Ground, ...?, happened, Whatever, Your, decom, Getting, Got, overlooked, Crack, )?, He, police, !?, happens, Suc, sets, what, Detective, GOT, Whatever, SET, Getting, Flying |
| 32 | glued, hid, melted, ., sent, breaths, etz, breath, Breath, baptized, watch, putting, tongues, braces, put, hid, bleach, icating, burying, aver, lifting, Illuminati, orneys, melting, withdrawing, numb, radios, inserts, amins, avert, breathing, puts, informants, lifting, hide, conscience, recommending, withdrawn, ransom, catch, Gael, Vern, roth, ears, Put, gins, breathed, attorneys, loss, biblical |
| 37 | hid, dropped, raided, fought, ungle, Hide, destruct, smuggled, abandoned, looted, attacked, barric, slid, dodged, drop, shut, drowned, hide, destruct, buggy, battled, shutdown, Hide, Attack, hid, rawl, inaccessible, avalanche, slipped, deleted, rawling, encrypted, withdrawn, Killed, dug, dropping, hoard, weapon, swallowed, defensive, destroy, exited, destroy, fight, Fighting, lost, deny, suppress, encrypt, aggressively |
| 45 | hopped, chats, pumped, paints, backed, spun, tread, coached, reefs, privately, noodles, buddies, malls, whisper, endorsements, squeezed, pals, blush, comed, edits, rallies, gigs, recol, mocked, curs, Bare, bubbles, warmed, chat, profiles, emails, Dreams, pads, chalk, interviewed, sneakers, rocked, Gloves, hubs, docs, shaved, Rise, primaries, listened, shy, essays, whispers, leeve, girlfriends, socks |

Table 8: Tokens with the largest $L_2$ norm of $\boldsymbol{P}$ entries for the COPA task. See Section C for more details.

| l# | Tokens $x$ with largest norm $\|\boldsymbol{P}_x\|_2$ |
|---|---|
| 0 | gression, rium, History, orer, aic, history, oration, ré, orative, amic, history, version, ural, osa, avage, ory, lia, range, History, rica, nation, root, USE, á, ination, ulation, mentation, issance, state, rum, adal, idden, jection, oly, ó, esis, orean, discovery, ria, ada, uration, entry, ord, verse, inations, ugal, itus, olics, ESSION, ativity |
| 5 | ., to, in, for, and, of, ,, s, the, be, 's, by, on, or, from, at, or, with, :, ly, a, an, ;, on, -, ., in, under, an, as, I, and, !, about, er, In, but, ?, A, is, ed, a, that, o, ers, S, ing, now, ), - |
| 10 | ., of, and, for, morph, votes, elector, with, uild, igraph, tatt, Assignment, as, contribut, advant, are, hod, Voters, matically, Init, rede, olon, on, rehabilit, neum, mog, looted, req, by, Claim, the, ynchron, dule, promot, socio, portfolios, goto, vulner, vote, setup, nominate, anism, s, subscrib, iop, lihood, slot, elist, ramid, ysc |
| 22 | in, In, in, be, In, .", being, Straw, -, its, a, Majority, of, a, Latest, the, Jack, ine, latest, it, Lawyers, Watts, "., "-, Massachusetts, their, .', been, ure, Till, '.", Signs, .'", Seventh, ?", Taxes, Atlanta, !", electric, at, IN, ide, Current, Ladies, KP, Jersey, Students, Knights, it, Anders |
| 32 | Se, Hum, Brazil, Mur, Hur, aver, Hum, Yugoslavia, Mour, jud, a, Hawai, Pag, Kant, ibal, Malaysia, EFF, Hur, .", adj, mur, Islam, and, Guinea, Britain, Sadd, Def, Niger, ,, Holland, amus, Hay, Ma, Appro, Mur, Countries, Wid, Asians, Nor, else, Calendar, Hed, Ved, ldom, english, Hind, mur, bury, Ded, hol |
| 37 | [SEP], +., Sk, Ble, Gre, cloud, Else, ., +,, "., uran, cs, Ever, 2048, Ble, Keefe, Hyp, athan, Lib, Fra, Exp, bro, Edit, Ros, Bean, Bo, Beck, Shell, sit, !., Saud, Phys, -, shell, Ol, BLIC, -, Over, ea, orthy, Shot, pn, pas, ester, Reviewed, Spe, sell, 2024 |
| 45 | Chance, Sw, chance, Nine, Shares, Chance, Scientists, Tw, Besides, Prof, chances, Sn, sw, TW, EFF, J, IJ, Besides, chance, Between, icist, GU, SW, pan, Ja, Psy, tw, Between, xon, Bj, Conj, Shares, Moh, UTH, Prediction, science, intend, Science, iov, Nine, jp, dds, NJ, Jr, y, Nin, etsy, Ibid, ymm, Reporting |

Table 9: Tokens with the largest $L_2$ norm of $\boldsymbol{P}$ entries for the RTE task. See Section C for more details.

| $l\#$ | Tokens $x$ with largest norm $||\boldsymbol{P}_x||_2$ |
|---|---|
| 0 | .", didn, doesn, don, "., ,", Didn, Doesn, didn, doesn, Don, wouldn, Wouldn, Does, ", couldn, DON, Did, hadn, But, Don, Isn, ).", DON, shouldn, ", Obviously, Obviously, Isn, don, hasn, )).,  Does, "?, ].", wasn, Did, ],", ,., Naturally, ...", ),", Would, ", But, ";, Naturally, ]., ,), DOES |
| 5 | ,, 't, !,, ,, ?,, didn, , not, to, +,, *,, ),, the, ., ],, considered, doesn, in, , ,), a, /,, ,[, you, don, ,,, ,., shouldn, (),, hasn, ;, for, thought, weren, hadn, thought, wasn, NOT, hair, ', .;, aren, ', Said, ,, couldn, isn, .—, idered |
| 10 | Shant, Georg, Expect, Led, Assistant, Amph, Registered, Ear, McA, THEIR, Prev, Emb, -,, Called, Gw, Alc, Until, Rhod, Introduced, that, Lat, Unt, Ul, Sv, Gh, to, of, Fernand, ,, elta, jac, unch, Ov, Sebast, apologised, JOHN, !"., Ll, hid, Somewhere, Been, Recently, and, Somebody, Fram, Coh, )., Sty, Elsewhere, Unt |
| 22 | 's, 're, A, 've, the, a, her, ', be, A, a, have, DOES, LIKE, ?", "?, 'd, )?, ?, s, ABOUT, ", Like, Pant, didnt, 'm, '?, E, The, doesnt, Was, re, :, ie, Surely, 'll, Corinth, At, Across, your, their, ?,, THEY, ...?, or, Fra, HOW, )/ |
| 32 | I, '?, he, "?, ?', )?, 't, "., .?, ...?, .", ?"., ':, He, !?, ?,, He, I, and, ?'", ?!", ?", ?)., !', he, .:, ?!, she, +., )!, '., .', !?", ,", ')., ???, !., ).", we, CLOSE, '., "!, .], .–, ????, '/, 're, .'" |
| 37 | ., 's, 't, ?, ', :, I, -, ', ,, of, he, B, B, in, I, and, 'm, -, s, ", 'd, by, for, ;, b, on, you, !, ", He, to, /, 've, y, 're, ed, with, ., 'll, a, back, the, b, she, He, E, C |
| 45 | 't, not, NOT, the, not, Not, never, Not, 's, 're, NOT, 've, you, of, [SEP], t, nt, Never, The, in, NEVER, he, to, the, [CLS], hardly, never, neither, I, ,, 'm, cannot, no, The, annot, it, Their, me, didnt, He, and, doesnt, Ear, a, ., Never, none, if, on, nobody |

Table 10: Tokens with the largest $L_2$ norm of $\boldsymbol{P}$ entries for the CB task. See Section C for more details.

